# Generalized Stacking Fault Energy of {10-11}<11-23> Slip System in Mg-Based Binary Alloys: A First Principles Study

**DOI:** 10.3390/ma12091548

**Published:** 2019-05-11

**Authors:** Yuchen Dou, Hong Luo, Jing Zhang, Xiaohua Tang

**Affiliations:** 1College of Materials Science and Engineering, Sichuan University of Science and Engineering, Zigong 643000, China; Luohong28@163.com (H.L.); Ttxh66@163.com (X.T.); 2College of Materials Science and Engineering, Chongqing University, Chongqing 400044, China; 3National Engineering Research Center for Magnesium Alloys, Chongqing 400044, China

**Keywords:** magnesium alloys, first-principle calculation, stacking-fault energy, ductility

## Abstract

In this work, the generalized stacking fault energies (GSFEs) of {10-11}<11-23> slip system in a wide range of Mg-X (X = Ag, Al, Bi, Ca, Dy, Er, Gd, Ho, Li, Lu, Mn, Nd, Pb, Sc, Sm, Sn, Y, Yb, Zn and Zr) binary alloys has been studied. The doping concentration in the doping plane and the Mg-X system is 12.5 at.% and 1.79 at.%, respectively. Two slip modes (slip mode I and II) were considered. For pure magnesium, these two slip modes are equivalent to each other. However, substituting a solute atom into the magnesium matrix will cause different effects on these two slip modes. Based on the calculated GSFEs, two design maps were constructed to predict solute effects on the behavior of the {10-11}<11-23> dislocations. The design maps suggest that the addition of Ag, Al, Ca, Dy, Er, Gd, Ho, Lu, Nd, Sm, Y, Yb and Zn could facilitate the {10-11}<11-23> dislocations.

## 1. Introduction

Magnesium alloys show great potential for application in the automotive and aerospace industry. However, the use of magnesium alloys is limited by their poor ductility at ambient temperatures [1,2,3,4,5]. In hexagonal close-packed (HCP) alloys, the softest deformation mode is the basal slip, which provides only two independent slip systems. This is far from the von Mises criterion, in which five independent deformation modes are required for a compatible deformation [6]. Furthermore, basal slips cannot accommodate the strain along the c axis.

Fundamentally, the alloy effects on the stacking fault energy (SFE) are related to electron localization morphology [7]. Many authors have simulated the SFEs of binary Mg-X alloys [7,8,9,10,11,12,13,14,15,16,17,18]. These authors mainly focus on the {0001}<11-20>, {10-10}<11-20> and {10-12}<11-23> slip systems. However, the data for the {10-11}<11-23> slip system is very limited.

It is widely accepted that the activation of <c+a> dislocations could improve the ductility of magnesium alloys. During a deformation process, <c+a> dislocations prefer to transform into low-energy structures [1,2,19,20]. The dissociated <c+a> dislocations could be immobile or mobile. It should be noted that the immobile transformation is the origin of high hardening and low ductility of magnesium alloys [1,2,18,21,22]. 

Wu et al. suggested that the key factor to enhance the ductility of magnesium alloys is increasing the <c+a> cross-slip (between the {10-11} and {11-22} planes) and multiplication rates to a level much faster than the immobile <c+a> transformation [2,21]. Anil et al. demonstrated that the {10-11}<11-23> dislocation could transform into a mobile structure as the following [19]:
1/3[11-23] → 1/6[20-23] + 1/6[02-23](1)

To this end, there must exist at least one stable stacking fault energy (SFE) on the {10-11} plane. The SFE for this dissociation could be searched out by the in-plane-relaxing permitted first principles simulations [19].

Ding et al. investigated the effects of Y on the generalized stacking fault energies (GSFEs) of the {10-11}<11-23> slip system [23]. The results illustrated that the stable SFE disappeared in the MgY alloy. In other words, the stable stacking fault on the {10-11} plane is not stable anymore. Thus, the addition of Y should promote the immobile transformation of the {10-11}<11-23> dislocation and inhibit the cross-slip between the {10-11} and {11-22} planes. However, in the Mg97Y3 (wt.%) alloys, the high-frequency <c+a> dislocations switching between the {10-11} and {11-22} planes were observed [2]. In other words, <c+a> dislocation on the {10-11} plane could cross slip onto the {11-22} plane and vice versa. This means the mobile {10-11}<11-23> transformation is preferred in the Mg97Y3 (wt.%) alloys.

We believe this discrepancy arises from the insufficient consideration of simulation modes. As shown in Figure 1a,b, when the shear stress is applied along the <11-23> direction, there should exist two deformation modes. In practice, the weaker one should be activated. For pure magnesium, these two slip modes are equivalent. However, if a solute atom is substituted into the fourth lattice plane, slip mode I and II should exhibit different GSFEs. To the best of our knowledge, previous studies have not addressed this issue yet. In this work, the GSFEs of {10-11}<11-23> slip system in a wide range of Mg-X (X = Ag, Al, Bi, Ca, Dy, Er, Gd, Ho, Li, Lu, Mn, Nd, Pb, Sc, Sm, Sn, Y, Yb, Zn and Zr) binary alloys will be studied, aiming to provide a basis for the design of high-performance magnesium alloys. It should be noted the selected alloying elements have a maximum solubility of >1.0 at.%.

## 2. Computational Details

First principle simulations in this work were carried out by ABINIT code [24,25], accompanied by the projector augmented wave method (PAW) [26]. The exchange-correlation was described by the generalized gradient approximation (GGA) in the Perdew-Burke-Ernzerhof (PBE) form [27]. The Brillouin zone was sampled using a 4 × 7 × 3 with Monkhorst–Pack method [28]. The cut-off energy was set as 40 Ha and the total energy converged within 4 × 10^−6^ Ha. A very recent study revealed that van der Waals interactions play a significant role in the deformation of magnesium, especially for the slip systems with relatively large inter-plane space [29]. Thus, in our simulations, the GGA-PBE functional was corrected by Grimme’s DFT-D3 functional to address the long-range electron-electron correlations [30]. Atomic models and electronic structures were visualized with VESTA [31].

As illustrated in Figure 1a, a super-cell with 56 atoms were constructed to calculate the GSFEs. A solute atom was substituted into the fourth plane to simulate the Mg55X1 systems. A vacuum width of 15 Å is added to avoid the interactions arising from the periodic images. For the slip mode I, the upper half of the cell was gradually displaced along the 1/3[11-23] direction, the displacement step is 0.05b (where b is the Burgers vector 1/3<11-23>). While, for the slip mode II, the lower half of the cell was gradually displaced along the 1/3[-1-12-3] direction. In all simulations, atoms were fully relaxed along the z and x directions. The calculated GSFEs curve for pure magnesium is shown in Figure 2; The stable and unstable SFEs for Mg and Mg55X1 are listed Table 1.

## 3. Results and Discussions

As shown in Figure 2, there exist two unstable SFEs and one stable SFE on the {10-11}<11-23> GSFE curve of pure magnesium. However, Yin et al. demonstrated that there exist three stable points (SFE1, SFE2 and SFE3) on the {10-11} γ surface [18,32]. The authors believe this is caused by different relaxing procedures. To cover the whole γ surface, the relaxation along the x direction must be forbidden. In this case, the in-plane stress cannot be released, which will result in “wrong” GSFEs [15]. Paradoxically, if the relaxation along the x direction is permitted, the upper (or lower) half of the cell could relax into a local minimum. In other words, the GSFEs will fail to cover the whole γ surface if the in-plane-relaxing is permitted.

The unstable SFE1, stable SFE and unstable SFE2 locate at 0.3b, 0.4b and 0.7b (where b is the Burgers vector 1/3<11-23>). The values for the GSFE0.3b, GSFE0.4b and GSFE0.7b are 175, 165 and 318 mJ m^−2^, respectively. The positions and values coincide with the data calculated with the same relaxing procedures (174, 163 and 315 mJ m^−2^ for GSFE0.3b, GSFE0.4b and GSFE0.7b, respectively) [23]. In that work, the van der Waals forces were calculated as well. Without consideration of the van der Waals forces, the positions of these three GSFEs remain unchanged but the values for SFE1, stable SFE and unstable SFE2 are much bigger (230, 185 and 395 mJ m^−2^ for GSFE0.3b, GSFE0.4b and GSFE0.7b, respectively) [19]. 

Let us continue our discussions with the Mg55Y1 alloy. The electronic structure of Mg55Y1 is shown in Figure 1c,d. In these figures, two periodic images are shown. Figure 1c,d reveal that there are more electrons between the fourth and fifth layers than between the third and fourth layers. This makes the slip mode I and II exhibit different GSFEs.

As listed in Table 1, for the slip mode I, the addition of Y decreases the GSFE0.3b from 175 to 147 mJ m^−2^; but increase the GSFE0.4b and GSFE0.7b from 164 and 318 mJ m^−2^ to 173 and 339 mJ m^−2^, respectively. In this case, GSFE0.4b is larger than that of GSFE0.3b. In other words, the GSFE0.4b is not a local minimum (stable point) anymore. Consequently, the immobile {10-11}<11-23> transformation is preferred under slip mode I. Interestingly, for the slip mode II, the addition of Y increase the GSFE0.3b from 175 to 185 mJ m^−2^; but decrease the GSFE0.4b and GSFE0.7b from 164 and 318 mJ m^−2^ to 117 and 291 mJ m^−2^, respectively. 

Decreased GSFE0.7b means that the nucleation of the {10-11}<11-23> dislocations could get easier [33]. The stability of a metastable state is determined by the energy difference between the peak and trough. So, we suggest that a larger value of GSFE0.3b-GSFE0.4b might increase the stability of the stacking fault on the {10-11} planes and promote the mobile <c+a> transformation. To get a global view of solute effects on the behavior of the {10-11}<11-23> dislocations, two “design” maps were constructed, as shown in Figure 3. In these maps, the GSFE0.7b and the values of GSFE0.3b-GSFE0.4b are involved. From left to right the probability of the mobile <c+a> transformation gets increased, corresponding with high ductility. Alternatively, a relatively lower value of GSFE0.3b-GSFE0.4b might promote the immobile transformation. From top to bottom, the nucleation of the {10-11}<11-23> dislocations could get easier. It should be emphasized again the high-frequency <c+a> dislocation switching between the {10-11} and {11-22} planes is the key factor of high ductility [2]. To this end, the dissociated {10-11}<11-23> dislocations must be mobile.

In pure Mg, the value of GSFE0.3b-GSFE0.4b is relatively small (11 mJ m^−2^). This means that the stability of a {10-11} stacking fault could be relatively lower. In other words, {10-11}<11-23> dislocations could transform into immobile structures or slip onto the {11-22} planes [33]. In Mg55Y1 alloy, the value of GSFE0.3b-GSFE0.4b is 68 mJ m^−2^ for slip mode II. In this case, the probability of the mobile <c+a> transformation gets increased. This coincides with experimental results that the high ductility of Mg97Y3 (wt.%) alloy is facilitated by the high-frequency <c+a> dislocation switching between the {10-11} and {11-22} planes [2].

The values of GSFE0.3b-GSFE0.4b for Mg55X1 (X = Dy, Er and Ho) are 66, 64 and 65 mJ m^−2^ under slip mode II, respectively. Experimentally, Sandlȍbes et al. demonstrated that the addition of 3 wt.% of these elements into pure Mg could increase the ductility in a great scale [34]. Under both slip mode I and II, Bi, Pb and Sn decrease GSFE0.7b dramatically. Moreover, our previous study revealed that Bi, Pb and Sn also decrease the maximum GSFE of {11-22}<11-23> slip systems obviously [17]. But, to the best of our knowledge, there is no evidence that Bi, Pb and Sn can dramatically increase the ductility of magnesium alloys. The authors suggest this is because, for both slip mode I and II, the values of GSFE0.3b-GSFE0.4b for Mg55X1 (X = Bi, Pb and Sn) are very small. Mg55Mn1 exhibits the largest value of GSFE0.3b-GSFE0.4b under slip mode I. Mg55Sc1 has a large value of GSFE0.3b-GSFE0.4b comparable to the value of Mg55ErY. However, the values of GSFE0.7b get increased by 38 and 35 mJ m^−2^ for Mg55Mn1 and Mg55Sc1, respectively. This means the nucleation of the {10-11}<11-23> dislocations could be relatively hard [33]. Under slip mode I, the values of GSFE0.3b-GSFE0.4b for Mg55X1 (X = Ag, Al and Zn) are 47, 34 and 41 mJ m^−2^, respectively. These three elements decrease GSFE0.7b by 15, 18 and 25, respectively. However, Wu et al. demonstrated that the activation energy barriers for the {11-22}<11-23> → {10-11} <11-23> cross-slip are relatively large [2].

## 4. Conclusions

In summary, solute effects on the GSFEs of {10-11}<11-23> slip system were studied by means of first principle calculations. Generalized stacking fault energy (GSFEs) of {10-11}<11-23> slip system in a wide range of Mg-X (X = Ag, Al, Bi, Ca, Dy, Er, Gd, Ho, Li, Lu, Mn, Nd, Pb, Sc, Sm, Sn, Y, Yb, Zn and Zr) binary alloys were studied. The doping concentration in the doping plane and the Mg-X system is 12.5 at.% and 1.79 at.%, respectively. Two slip modes (slip mode I and II) were considered. For pure magnesium, these two slip modes were equal to each other. However, substituting a solute atom into the magnesium matrix will cause different effects on these two slip modes, as shown in Figure 2. These findings are also applicable to the GSFEs calculation when the slip plane exhibits a zigzag feature (such as {10-11} plane in Mg, Zr and Ti). Two design maps, as shown in Figure 3, were constructed to predict solute effects on the behavior of the {10-11}<11-23> dislocations. From left to right, the stability of the mobile structure or the probability of the mobile <c+a> transformation could get increased. From top to bottom, the nucleation of the {10-11}<11-23> dislocations could get easier. The design map suggests that the addition of Ag, Al, Ca, Dy, Er, Gd, Ho, Lu, Nd, Sm, Y, Yb and Zn could facilitate the {10-11}<11-23> dislocations. The addition of Mn, Sc and Zr increases the GSFE0.7b dramatically. Thus, the nucleation of the {10-11}<11-23> dislocations could get harder in Mg55Mn1, Mg55Sc1 and Mg55Zr1. In Mg55Bi1, Mg55Pb1 and Mg55Sn1, the {10-11}<11-23> dislocation prefers to transform into immobile structures.

## Figures and Tables

**Figure 1 materials-12-01548-f001:**
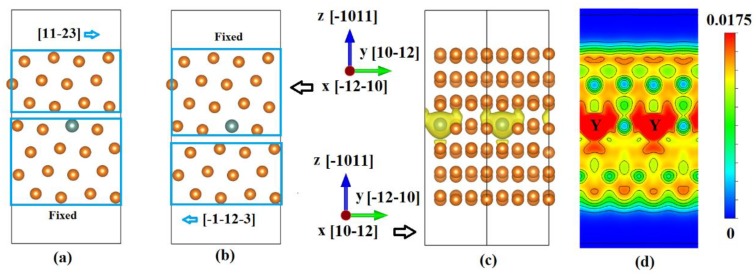
Atomic configurations for the {10-11}<11-23> slip system. A solute atom is substituted into the fourth plane for the Mg55X1 systems. The iso-surface level is 0.0175 a0^−3^ (a0: Bohr radius). (**a**) Slip mode I. The lower four layers of atoms are fixed; and the upper three layers of atoms are displaced along the 1/3[11-23] direction. (**b**) Slip mode II. The upper four layers of atoms are fixed; and the lower three layers of atoms are displaced along the 1/3[-1-12-3] direction. (**c**) Electronic structures of the Mg55Y1. Two periodic images are shown. The iso-surface level is 0.0175 a_0_^−3^ (a_0_ is Bohr radius). (**d**) A slice view of electronic structure of the Mg55Y1. Two periodic images are shown.

**Figure 2 materials-12-01548-f002:**
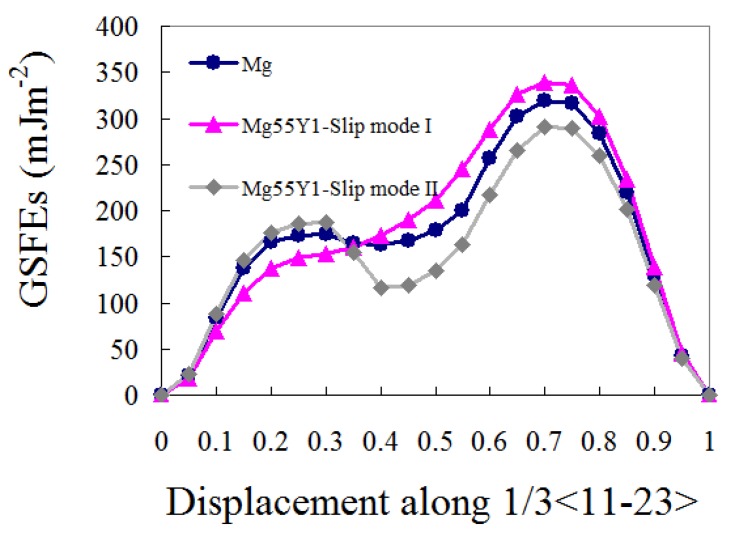
Calculated {10-11}<11-23> generalized stacking fault energies (GSFEs) for pure magnesium and Mg55Y1. For pure magnesium, slip mode I is equal to slip mode II. However, substituting a solute atom into the magnesium matrix will cause different effects on these two slip modes. Taking Mg55Y1 as an example, under slip mode I, the stable SFE has disappeared.

**Figure 3 materials-12-01548-f003:**
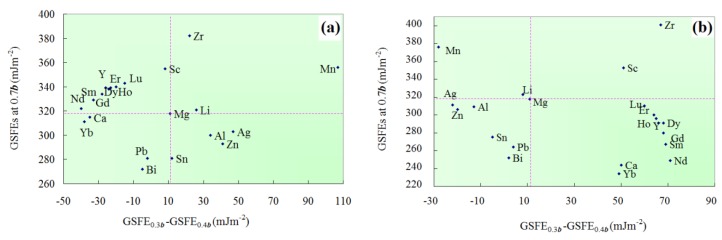
Alloy effects on GSFE0.7 b and GSFE0.3b-GSFE0.4b. (**a**) Slip mode I. (**b**) Slip mode II. From left to right the probability of the mobile <c+a> transformation could get increased. From top to bottom, the nucleation of the {10-11}<11-23> dislocations could get easier.

**Table 1 materials-12-01548-t001:** The calculated GSFEs (mJ m^−2^) for pure Mg and Mg55X1 systems. For pure Mg, GSFE0.4b is stable (where b is the Burgers vector 1/3<11-23>). If the value of GSFE0.4b is bigger than GSFE0.3b, GSFE0.4b is not a local minimum (stable point) anymore, as shown in Figure 2.

	Slip Mode I			Slip Mode II		
	0.3b	0.4b	0.7b	0.3b	0.4b	0.7b
Mg56	175 174 a 230 b	164 163 a 185 b	318 315 a 395 b	175	164	318
Mg55Ag1	185	138	303	166	188	311
Mg55Al1	181	147	300	163	176	309
Mg55Bi1	141	146	272	144	142	252
Mg55Ca1	130	165	315	171	121	244
Mg55Dy1	146	170	338	185	119	291
Mg55Er1	151	171	340	186	122	300
Mg55Gd1	140	168	334	183	115	280
Mg55Ho1	148	171	339	186	121	296
Mg55Li1	176	150	321	173	165	323
Mg55Lu1	157	172	343	189	129	310
Mg55Mn1	242	135	356	182	210	376
Mg55Nd1	123	163	322	177	106	249
Mg55Pb1	150	152	281	150	146	264
Mg55Sc1	180	172	355	193	142	353
Mg55Sm1	133	166	329	180	111	267
Mg55Sn1	159	147	281	150	155	275
Mg55Y1	147~180 c	173~195 c	339~355 c	185	117	291
Mg55Yb1	122	160	311	170	121	234
Mg55Zn1	177	136	293	162	182	306
Mg55Zr1	201	179	382	207	140	401

^a^ Reference [23], the van der Waals forces were calculated; ^b^ Reference [19], the van der Waals forces were not calculated; ^c^ Reference [23], the van der Waals forces were calculated. In this reference, the alloy is Mg47Y1 but the solute concentration on the doped plane is equal to the present work.

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
