# Peer review of "Generalized Stacking Fault Energy of {10-11}<11-23> Slip System in Mg-Based Binary Alloys: A First Principles Study"

_materials, 2019, doi:10.3390/ma12091548_

Reviewer 1 Report

This paper investigates the effect of various doping elements on Generalized Stacking Fault Energy (GPFE) in binary Mg alloys by means of ab initio calculations. This study is focused on the {10-11}<11-23>pyramidal slip system, which, in contrast to the basal slip, is not systematically addressed the literature exploring the effect of alloying elements on stacking faults and dislocations in Mg. Overall, the manuscript provides some useful insights into mobility and nucleation of stacking faults and dislocations within the investigated slip mode and, therefore, it is worth publishing. However, I have a few comments that should be addressed before final acceptance.

- I think that it is worth mentioning in the abstract how many doping elements have been investigated and which of them were found to have the strongest effect on GSFE.

- I find the introduction a bit too short. It does not describe the results of previous (very similar and numerous) studies of doping elements on the GSFEs in the Mg-X systems. The authors should CITE and describe which slip systems have been already investigated and what are the main conclusions of these studies, e.g., of 

W.Y. Wang et al., Mater. Res. Lett. 2 (2014) 29-36

https://doi.org/10.1080/21663831.2013.858085

Q. Dong et al., J. Mater. Sci Technol 34 (2018) 1773-1780 

https://doi.org/10.1016/j.jmst.2018.02.009

Z. Pei et al., Phys. Rev. B 92 (2015) 064107

https://doi.org/10.1103/PhysRevB.92.064107

etc.

- In the introduction and discussion, the authors provide the doping concentration for  experiments from literature, but not for the investigated systems. It is worth indicating ( e.g., in the abstract and in the introduction) the doping concentration (in wt.% or at.%) in the doping plane itself and in the Mg-X system

- line 84:<10-11>is not a direction, it is a plane and should be indicated as {10-11}

- In Figure 1d, the scale bar and its units are not explained

- Figure 2 provides the computed GSFE curve for pure Mg where both slip modes are identical. In the text, the authors state that for the doped systems it is not the case but hey do not provide any GSFE curves to support these statement. Here it is worth providing at least one GSFE curve for a doped system (e.g., for Mg55Y, which is discussed in detail in the section 3, or for any other system where this difference is particularly prominent)

- In the capture of Figure 2: 

 (i)  I suggest to explicitly indicate the investigated slip system {10-11}<11-23>.  

 (ii) "Pure" -> "pure"

- In Table 1, it not clear what is 03b, 04b and 07b. I guess, it is a reaction coordinate along the displacement vector. These notations should be better explained in the Table capture and in the text, e.g., line 102: "(Burgers vector)" -> (where b is the Burgers vector 1/3<11-23>)

-In the text, the GSFE units are systematically provided as "mJm-2" (without superscript) 

- I find the conclusions of the article confusing for the readers. This section does not contain any figure, however the authors describe the main effect of doping elements "from left to right" and "from top to bottom". Here, one should clearly describe which properties are varied in this directions. Besides, the authors should indicate which alloying elements have been investigated, in which concentration;  which of the investigated elements can be expected to have a particularly strong effect on dislocation properties, and which will rather have no prominent effect; what is the possible explanation of the observed effect of doping elements on computed GSFE: atomic radius, electronic structure, bond strength, binding energy, etc. ? 

 - In the reference list

 (i) the names of several authors appear incorrectly (e.g., see "?" symbol) in refs 2, 4, 24

 (ii)orshould be replaced within refs 7, 8, 9, 11, 12 

Author Response

Dear Madam or Sir,

Thank you so much for your constructive advices on my manuscript. We have read your advices very carefully. Many changes, colored in green, have been made in the revised manuscript.

 This paper investigates the effect of various doping elements on Generalized Stacking Fault Energy (GPFE) in binary Mg alloys by means of ab initio calculations. This study is focused on the {10-11}<11-23>pyramidal slip system, which, in contrast to the basal slip, is not systematically addressed the literature exploring the effect of alloying elements on stacking faults and dislocations in Mg. Overall, the manuscript provides some useful insights into mobility and nucleation of stacking faults and dislocations within the investigated slip mode and, therefore, it is worth publishing. However, I have a few comments that should be addressed before final acceptance.

 - I think that it is worth mentioning in the abstract how many doping elements have been investigated and which of them were found to have the strongest effect on GSFE.

Answer: The doping elements were listed in the abstract. Page 1 line 13-16.

       Since there exist two slip modes and each mode has three values 03b, 0.4b and 0.7b, it is difficult to list the alloying effect in the abstract. However, we added the following statement in the abstract

“The design maps suggest that the addition of Ag, Al, Ca, Dy, Er, Gd, Ho, Lu, Nd, Sm, Y, Yb and Zn could facilitate the {10-11}<11-23>dislocations.”

- I find the introduction a bit too short. It does not describe the results of previous (very similar and numerous) studies of doping elements on the GSFEs in the Mg-X systems. The authors should CITE and describe which slip systems have been already investigated and what are the main conclusions of these studies, e.g., of 

W.Y. Wang et al., Mater. Res. Lett. 2 (2014) 29-36

https://doi.org/10.1080/21663831.2013.858085

Q. Dong et al., J. Mater. Sci Technol 34 (2018) 1773-1780 

https://doi.org/10.1016/j.jmst.2018.02.009

Z. Pei et al., Phys. Rev. B 92 (2015) 064107

https://doi.org/10.1103/PhysRevB.92.064107

etc.

Answer: We added a new paragraph to describe the previous studies. Page 1 line 31-34. The mentioned and many other references were cited

 In the introduction and discussion, the authors provide the doping concentration for  experiments from literature, but not for the investigated systems. It is worth indicating ( e.g., in the abstract and in the introduction) the doping concentration (in wt.% or at.%) in the doping plane itself and in the Mg-X system

Answer: The following statement was added in the abstract.

“The doping concentration in the doping plane and the Mg-X system is 12.5 at.% and 1.79 at.%, respectively. “

- line 84:<10-11>is not a direction, it is a plane and should be indicated as {10-11}

Answer:<10-11>  was changed into {10-11} page 4 line 132

- In Figure 1d, the scale bar and its units are not explained

Answer: The following statement was added in captain of Figure 1.

“The iso-surface level is 0.0175 a0-3 (a0 is Bohr radius).”

- Figure 2 provides the computed GSFE curve for pure Mg where both slip modes are identical. In the text, the authors state that for the doped systems it is not the case butheydo not provide anyGSFE curves to supportthese statement. Here it is worth providing at least one GSFE curve for a doped system (e.g., for Mg55Y,which is discussedin detail in the section 3, or for any other system where this difference isparticularly prominent)

Answer: Figure 2 was updated. In the revised figure,the curves for Mg55Y1 were added.

- In the capture of Figure 2: 

 (i)  I suggestto explicitly indicatethe investigated slip system {10-11}<11-23>.  

 (ii) "Pure" -> "pure"

Answer: {10-11}<11-23>was addedand  "Pure" haswas changedinto "pure"

- In Table 1, it not clear what is 03b, 04b and 07b. I guess, it is a reaction coordinate along the displacement vector. These notations shouldbe better explainedin the Table capture and in the text, e.g., line 102: "(Burgers vector)" -> (where b is the Burgers vector 1/3<11-23>)

Answer: 03b, 04b and 07b have beenexplanationin the Table capture and in the text (line page 4 line 140)

-In the text, the GSFE unitsare systematically providedas "mJm-2" (without superscript) 

Answer: "mJm-2" was changedinto "mJm-2"

- I find the conclusions of the article confusing for the readers. This sectiondoes not contain anyfigure, however the authors describe the main effect of doping elements "from left to right" and "from top to bottom". Here, one shouldclearly describewhich properties are variedinthisdirections. Besides, the authors shouldindicatewhich alloying elements have been investigated, in which concentration; which of the investigated elements can be expectedto have aparticularly strong effect on dislocation properties, and which will rather have no prominent effect; whatis the possible explanation ofthe observed effect of doping elements on computed GSFE: atomic radius, electronic structure, bond strength, binding energy, etc. ? 

Answer: 1. Conclusionswere updated. Now, the readers could find “the Figures” easily.

2. Alloying elements and their concentrations were added into the conclusions.

3. The following statementswere addedto describe the alloying effects.

“The design map suggest that the addition of Ag, Al, Ca, Dy, Er, Gd, Ho, Lu, Nd, Sm, Y, Yb and Zn could facilitate the{10-11}<11-23>dislocations. The addition of Mn, Sc and Zr increases the GSFE0.7b dramatically. Thus, the nucleation of the {10-11}<11-23>dislocations could get harder in Mg55Mn1, Mg55Sc1 and Mg55Zr1. In Mg55Bi1, Mg55Pb1 and Mg55Sn1, the {10-11}<11-23>dislocation prefer to transform into immobile structures.

4.Fundamentally, thealloy effects on the SFEare relatedto the electron localization morphology.  Page 1 line 31.

“Wang, William Yi, Shun Li Shang, Yi Wang, Zhi-Gang Mei, Kristopher A. Darling, Laszlo J. Kecskes, Suveen N. Mathaudhu, Xi Dong Hui, and Zi-Kui Liu. "Effects of Alloying Elements on Stacking Fault Energies and Electronic Structures of Binary Mg Alloys: A First-Principles Study." Materials Research Letters 2, no. 1 (2014): 29-36.”

- In the reference list

 (i) the names of several authors appear incorrectly (e.g., see "?" symbol) inrefs2, 4, 24

 (ii)orshould be replacedwithinrefs7, 8, 9, 11, 12 

Answer:1. The name of authors“ Stefanie Sandlöbes, Sören Müller, Sandlöbes, S”were correct.

2.orwere changed into the right style.

 Thank you again for your help.

Best wishes,

Yuchen Dou

Reviewer 2 Report

In this paper, authors used Ab initio calculation to obtain the generalized stacking energy surface of pure Magnesium alloys and of Mg-X binary alloys.

The first question is: how did authors choose the binary mixtures? Did they based this study on a previous study? 

 In the introduction please describe better theand the MgY (3wt%) notations.

In the computational details, how big is the displacement of the cells? Did authors applied a constant force to the cell, or they built several different conformations including the displacement? 

In the conclusions section I suggest to authors to give a very brief summary of the main results regarding the binary systems that should be available to readers, especially to experimentalists.

Did authors calculated for the binary mixtures only the 0.3 ,0.4 and 0.7 points, or the all curve as the one shown figure 1? It should be interesting to see if there are any differences in the trend of the GSFEs using different models. This analysis could be done just for one or two interesting binaries and not for all to keep the text clear.

Author Response

Dear Madam or Sir,

Thank you so much for your constructiveadviceson my manuscript. We have read youradvicesvery carefully.Many changes, colored in green, have been madein the revised manuscript.

 The first question is: how did authors choose the binary mixtures? Did they based this study on a previous study? 

Answer: “the selected alloying elements have a maximum solubility of >1.0 at.%. “ was added. Page 2 line 63.

 In the introduction please describe better theand theMgY(3wt%) notations.

Answer:1.MgY(3wt%) was changed into Mg97Y3 (wt%).

2. Sincedislocations exist on both {10-11} and {11-22} planes,the following statement was addedto avoid confusion. Page 2 line 53-54

“In other words,dislocation on the {10-11} plane could cross slip onto the {11-22} plane, and vice versa.

In the computational details, how big is the displacement of the cells? Did authors applied a constant force to the cell, or they builtseveral differentconformations including the displacement? 

Answer: 1.“The displacement step is 0.05b (where b is the Burgers vector 1/3<11-23>)” was added. Page 2 line 80.

2.In the simulations,several differentsupercellswere constructed to include the displacement.

 In the conclusions section I suggest to authors to give avery brief summary of the main results regarding the binary systems that should be available to readers, especially to experimentalists.

Answer:The conclusions were updated. In the revised conclusions,a brief summary of the main results regarding the binary systems were added. Page 6 line 212-216.

 Did authors calculated for the binary mixtures only the 0.3 ,0.4 and 0.7 points, or the all curve as the oneshownfigure 1? It shouldbe interestingto see if there are any differences in the trend of the GSFEs using different models.This analysis could be donejust for one or two interesting binaries and not for all to keep the text clear.

Answer: Figure 2 was updated. In the revised figure,the curves for Mg55Y1 were added.

Thank you again for your help.

Best wishes,

Yuchen Dou

Round  2

Reviewer 1 Report

The authors have answered all the questions and the manuscript has been improved. I recommend the publication in the current form.